# Clinical efficacy analysis of the Chinese medicine Paiteling applied to human papillomavirus infection: A retrospective study

Na He[1]☯, Lin Song[2]☯, Zhaoxia Lu[2]☯, Qingsong Zeng[2], Wumei Jin[3], Wenrong He[2], Cunjian Yi[2,4]*

1 Department of General, The First Affiliated Hospital of Yangtze University, Jingzhou, Hubei, China,
2 Department of Obstetrics and Gynecology, The First Affiliated Hospital of Yangtze University, Jingzhou, Hubei, China, 3 Department of Oncology, The First Affiliated Hospital of Yangtze University, Jingzhou, Hubei, China, 4 Department of Clinical Research Center for Cancer, The First Affiliated Hospital of Yangtze University, Jingzhou, Hubei, China

☯ These authors contributed equally to this work.
* cunjiany@yangtzeu.edu.cn

**Data Availability Statement:** All relevant data are within the paper and its Supporting information files.

## Abstract

### Aim

To investigate the clinical efficacy of population-based treatment of human papillomavirus (HPV) infections with Paiteling.

### Methods

Between 1 June 2024 and 31 August 2024, 575 HPV-infected patients attending The First People's Hospital of Jingzhou gynecology outpatient clinic from October 2020 to December 2023 were retrospectively collected, all of whom were analyzed for HPV subtype infection and the presence or absence of cytomorphological abnormality using HPV-DNA testing and TCT; they were divided into 319 cases in the Paiteling group and 256 cases in the Interferon group, and the patients of both groups were subjected to TCT 1 month after the end of the administration of the medication and HPV review.

### Results

1. The three most frequent subtypes of HPV in this data are HPV 16, HPV 52, and HPV 58, all of which are high-risk types; 2. The results of the post-treatment review of HPV infection showed that the overall effective rate of the Paiteling group was better than that of the Interferon group, and the difference in the cumulative effective rate between the two groups was statistically significant (P < 0.001); 3. Comparative analysis of patients with 14 high-risk types of human papillomavirus in a single infection showed that the overall conversion rate in the Paiteling group was 76.3%, higher than that of the Interferon group, which was 36.7%. The difference was statistically significant (P < 0.001).

**Funding:** This study was supported by the Hubei Provincial Medical Leaders Cultivation Project Special Funding Program (E Health Planning and Health Development (2013) No. 4) to Cun-jian Yi. And supported by Jingzhou Joint Research Fund Project, Project No. 2024LHY22 to Qing-song Zeng.

**Competing interests:** The authors have declared that no competing interests exist.

## Conclusion

The overall efficacy of Paiteling for cervical HPV infection is significantly better than that of Interferon, and it is worth promoting its use in the clinic.

## 1. Introduction

Persistent human papillomavirus infection is the main cause of cervical cancer development. According to the Global Cancer Statistics Report 2020, cervical cancer is considered one of the fourth leading causes of cancer in women in terms of both morbidity and mortality [1]. According to the latest update of the World Health Organisation (WHO) carcinogens, some HR-HPVs are classified as Class I carcinogens, including HPV types 16, 18, 33, 31, 35, 39, 45, 51, 52, 56, 58, and 59 [2]. Generally speaking, HPV infection usually disappears naturally within a few months, and about 90% disappear within 2 years. However, a small number of people will continue to be infected with HPV, which will transform into cervical intraepithelial neoplasia and even develop into invasive cervical cancer [3]. In recent years, with the increasing incidence of cervical cancer and the trend towards younger age groups, early clearance of persistent HPV infection has become the primary goal of clinicians. Some patients with persistent HPV may develop cervical intraepithelial neoplasia, and interventions such as physiotherapy or electrosurgery may lead to structural damage to the cervix or increase psychological stress in young patients with reproductive needs. Therefore, it is particularly urgent to explore an effective drug treatment option. Because of its unique efficacy in antiviral therapy, the proprietary Chinese medicine Paiteling has been increasingly widely used in anti-HPV treatment.

Paiteling is a pure Chinese medicine preparation, developed by the Chinese Academy of Sciences in 1993, and its main ingredients are several Chinese herbs such as Da Qing Ye, Honeysuckle, Bitter Ginseng, Crow's Nest, and Snake Bed Seed. Da Qing Ye, Bitter Ginseng, and Honeysuckle can interfere with and block the synthesis of HPV-DNA and inhibit viral replication, at the same time, they have the effects of clearing heat and removing toxins, activating blood circulation and removing blood stasis, and improving and stimulating autoimmunity [4]; Both Serratia marcescens [5] and Crowberry [6] have the ability to exfoliate keratinocytes while removing and inhibiting HPV pathogens from proliferating cells; in other words, Paiteling allows the virus to be exfoliated along with the host cell before the epithelial cell completes the reproduction of the release, thus effectively removing the virus. Internationally recognized studies have shown that the action of Paiteling is limited to the basal layer of the skin and does not enter the dermis, causing no damage to the dermis or subcutaneous tissues, thus unifying its high potency and low toxicity [7]. At present, there is no clear drug or treatment to eradicate HPV infection in either the international guidelines or the domestic consensus of the relevant experts, so the search for drugs with good efficacy and low toxicity and side effects in clinical work is what clinicians have been exploring. This study aims to investigate the clinical efficacy analysis of the Chinese medicinal Paiteling and the traditional drug Interferon in the treatment of HPV infection.

## 2. Methods

### 2.1 Study subjects

Between 1 June 2024 and 31 August 2024, we retrospectively collected patients who attended the gynecology outpatient clinic of The First People's Hospital of Jingzhou from 1 October

2020 to 1 December 2023, all of whom underwent both thin-layer cytology (TCT) and human papillomavirus (HPV) DNA genotyping. Colposcopy was performed on patients who had medical indications for undergoing a colonoscopy, and biopsies were taken from abnormal sites under colposcopy for histopathological examination. Five hundred and seventy-five patients who met the inclusion criteria were screened and divided into 319 cases in the Paiteling group and 256 cases in the Interferon group. This study was approved by the Ethics Committee of the First People's Hospital of Jingzhou (Approval No. KY202434), which allowed access to the data used for this study.

## 2.2 Inclusion and exclusion criteria

Inclusion criteria: (1) Persistent HPV positivity and no cancer in colposcopic cervical biopsy; (2) history of sexual life; (3) no important organs and oncological diseases; (4) no gynecological examination, vaginal medication, vaginal douche within two weeks; (5) no recent oral antiviral medication and HPV vaccine; (6) negative infectious disease examination.

Exclusion criteria: (1) immunocompromised; (2) received local or systemic immune enhancer treatment in the past six months; (3) acute inflammation of the reproductive tract; (4) pregnancy and breastfeeding; (5) long-term use of immunosuppressants; (6) suffering from severe cardiac, pulmonary, hepatic and renal dysfunction; (7) received local or systemic antiviral treatment in the past two months; (8) do not accept the full treatment and follow-up.

## 2.3 Methods of drug administration

(1) Paiteling Group: Patients in the Paiteling Group were given vaginal medication by specially trained staff, using 3 days and then stopping for 4 days, stopping use during menstruation, with 7 days being a cycle, and a total of 6 cycles of use constituting a whole course of treatment. During the treatment period, patients were urged to strictly avoid sexual intercourse, tub baths, and other phenomena. (2) Interferon Group (Jinshuxi): Starting on the 3rd day after the patient's menstrual period, the drug was placed in the posterior fornix of the vagina every night before bedtime, 1 tablet each time, every other day, avoiding the menstrual period, and the treatment was carried out for three consecutive months. Follow-up with TCT and HPV was conducted 1 month after the end of the medication.

## 2.4 Evaluation of efficacy

(1) Ineffective: re-infection in which the original HPV subtype remains positive or is different from the previous subtype; (2) Partial conversion: reduction in the number of subtype species in multiple-mixed individuals; (3) Total conversion: negative results for the original HPV subtype.

## 2.5 Statistical methods

In this study, the data were analyzed and processed using the statistical software SPSS version 25.0. The Mann-Whitney U test was used to assess the age-related variance in the measurement data, and the chi-square test was used to compare the rates of categorical data. Results were considered statistically significant if $P < 0.05$.

## 3. Results

### 3.1 Comparison of the general conditions of the two groups

There were a total of 319 patients in the Paiteling group, aged 44 (34–53) years, of whom 186 were single HPV infected and 133 were multiple HPV infected, with no cancerous tissue

further confirmed on examination. In the Interferon group, there were 256 cases, aged 49 (38–56.5) years, including 184 cases of single HPV infection and 72 cases of multiple HPV infection; no cancer was found in the pathological examination.

## 3.2 The prevalence of the HPV genotypes

In the 575 samples, the total frequency of infection for each subtype of the infected persons was 887 times (the frequency of infection for double infected persons was recorded as 2 times, and so on), of which 84.7% (751/887) were high-risk types and 15.3% (136/887) were low-risk types. The three most frequently infected subtypes were HPV 16, HPV 52, and HPV 58, with 13.4% (119/887), 13.4% (119/887) and 10.4% (92/887) infections, respectively, as shown in Table 1.

**Table 1. Distribution of infection subtypes in HPV positive patients.**

| HPV genotypes | Person-time of infection | Constituent ratio* |
|---|---|---|
| **High-risk type** | | |
| 16 | 119 | 13.4 |
| 18 | 55 | 6.2 |
| 31 | 13 | 1.5 |
| 33 | 28 | 3.2 |
| 35 | 16 | 1.8 |
| 39 | 56 | 6.3 |
| 45 | 7 | 0.8 |
| 51 | 50 | 5.6 |
| 52 | 119 | 13.4 |
| 53 | 70 | 7.9 |
| 56 | 29 | 3.3 |
| 58 | 92 | 10.4 |
| 59 | 24 | 2.7 |
| 66 | 34 | 3.8 |
| 67 | 6 | 0.7 |
| 68 | 24 | 2.7 |
| 73 | 5 | 0.6 |
| 82 | 3 | 0.3 |
| 83 | 1 | 0.1 |
| **Low-risk type** | | |
| 6 | 15 | 1.7 |
| 11 | 9 | 1.0 |
| 40 | 15 | 1.7 |
| 42 | 24 | 2.7 |
| 43 | 10 | 1.1 |
| 44 | 16 | 1.8 |
| 54 | 15 | 1.7 |
| 55 | 12 | 1.4 |
| 61 | 8 | 0.9 |
| 81 | 12 | 1.4 |

HPV: human papillomavirus.

**Table 2. Comparison of treatment efficacy between the two groups [n (%)].**

| Groups | Total cases | single infection | multiple infection | | cumulative effective rate |
|---|---|---|---|---|---|
| | | | Partial conversion | Total conversion | |
| Paiteling | 319 | 142(76.3) | 56(42.1) | 70(52.6) | 268(84.0) |
| Interferon | 256 | 67(36.4) | 18(25.0) | 21(29.2) | 106(41.4) |
| *P-value*[a] | | <0.001 | <0.05 | 0.001 | <0.001 |
| $X^2$ | | 60.004 | 5.925 | 10.419 | 113.391 |

N: number of cases; %: rate.

[a]*P*-value was calculated by chi-square test.

### 3.3 Analysis of the efficacy of the two groups of drugs on HPV infection

Of the 575 patients with HPV infection, they were categorized as mono-infected and multi-infected according to the type of infection. In a single infection, the conversion rate in the Paiteling group was 76.3% (142/186), with significantly better efficacy than that in the Interferon group of 36.4% (67/184), and the difference was statistically significant (P<0.001). Among the multiple infections, they were further divided into partial conversion and total conversion according to the review results, and the difference between the two groups of total conversion of multiple infections was statistically significant (P = 0.001), and the conversion rate in the Paiteling group (52.6%) was higher than that in the Interferon group (29.2%); Among them, patients with partial conversion of multiple infections were slightly higher in the Paiteling group (42.1%) than in the Interferon group (25.0%, 17/69), and the difference in efficacy between the two groups was statistically significant (P< 0.05). In terms of cumulative effective rate, the Paiteling group (84%) was significantly higher than the Interferon group (41.4%), and the difference in efficacy was statistically significant (P < 0.001) (Table 2).

### 3.4 Analysis of the efficacy of the two groups of drugs in the rate of conversion to 14 HPV mono-infections

Nucleic acid testing for 14 high-risk HPV subtypes (HPV 16, 18, 31, 33, 35, 39, 45, 51, 52, 56, 58, 59, 66, and 68) is recommended in the WHO guidelines for cervical cancer screening and pre-cancerous lesions issued in 2021, as well as in the Recommended Methods for China's National Health and Wellness Cervical Cancer Screening Programme [8]. Fourteen high-risk HPV subtypes were classified from 318 patients with mono-infection in the Paiteling group and 182 patients with mono-infection in the Interferon group, and the conversion rates of the 2 drugs for each HPV subtype were compared and analyzed. Eight HPV subtypes converted ≥80.0% in the treatment of the Paiteling group, whereas only two HPV33 and 45 converted ≥80.0% in the Interferon. The top 3 subtypes of highly prevalent HPV in this study were 16, 52, and 58, and the conversion rates were 75.5%, 80.6%, and 52.9% in the Paiteling group, and 24.0%, 38.1%, and 23.1% in the Interferon group, respectively, with statistically significant differences in the efficacy of the three groups of HPV 16, 18, and 52 (P < 0.05). In terms of the overall efficacy of the 2 groups, the conversion rate was 76.3% in the Paiteling group and 36.7% in the Interferon group, p<0.001, which is statistically significant, indicating that the overall efficacy of the Paiteling group was significantly better than that of the Interferon group in the treatment of the 14 high-risk types of HPV infections. Currently, HPV is classified into more than 200 subtypes by molecular biology technology, of which HPV16 and 18 are the most common high-risk types of cervical cancer, and the efficacy of Patulin against HPV16 and 18 is significantly higher than that of interferon. See Table 3.

**Table 3. Analysis of the efficacy of the two groups of drugs on the rate of conversion to 14 HPVs [n (%)].**

| Subtype | Paiteling Group | | Interferon Group | | $X^2$ | P-value[a] |
|---|---|---|---|---|---|---|
| | Total cases | Conversion rate | Total cases | Conversion rate | | |
| **HPV** | | | | | | |
| 16 | 53 | 40(75.5) | 25 | 6(24.0) | 18.601 | <0.001 |
| 18 | 18 | 11(61.1) | 13 | 3(23.1) | 4.409 | 0.036 |
| 31 | 5 | 4(80.0) | 2 | 2(100.0) | 0.467 | 1 |
| 33 | 8 | 6(75.0) | 3 | 0(0.00) | 4.950 | 0.061 |
| 35 | - | - | 5 | 3(60.0) | - | - |
| 39 | 11 | 9(81.8) | 11 | 7(63.6) | 0.917 | 0.635 |
| 45 | 1 | 1(100) | 1 | 1(100) | - | - |
| 51 | 5 | 5(100) | 15 | 7(46.7) | 4.444 | 0.055 |
| 52 | 31 | 25(80.6) | 21 | 8(38.1) | 9.775 | 0.002 |
| 56 | 5 | 5(100) | 5 | 1(20.0) | 6.667 | 0.048 |
| 58 | 17 | 9(52.9) | 25 | 7(28.0) | 2.669 | 0.102 |
| 59 | 6 | 5(83.3) | 4 | 3(75.0) | 0.104 | 1 |
| 66 | 6 | 5(83.3) | 5 | 2(40.0) | 2.213 | 0.242 |
| 68 | 7 | 7(100) | 4 | 1(25.0) | 7.219 | 0.024 |
| Total | 173 | 132(76.3) | 139 | 51(36.7) | 9.864 | <0.001 |

HPV: human papillomavirus.

[a]P-value was calculated either by chi-square test or Fisher's exact test.

## 3.5 Adverse reactions

The incidence of adverse reactions was higher in the Paiteling group than in the Interferon group, and several different side effects may occur during the same patient's dosing period. For example, patients may experience symptoms such as localized burning sensation and Shedding of cervical epidermal cells simultaneously. In contrast, only 18 patients in the interferon group reported an increase in conscious vaginal discharge. The specific data are detailed in Table 4.

## 4. Discussion

In this retrospective study, our epidemiologic research found that the top three HPV incidence rates were for HPV types 16, 52, and 58, all of which are high-risk types. This finding is largely consistent with the large-sample epidemiologic survey conducted in Northwest Eurasia, China [9]. In this study, from the analysis of the cumulative effective rate situation, the cumulative effective rate of HPV mono-infection and multi-infection patients using Paiteling was 84.0%,

**Table 4. Comparison of the occurrence of adverse reactions in the two groups [n (%)].**

| Groups | Total cases | Increased vaginal secretions | Shedding of cervical epidermal cells | Localized burning sensation | Lower abdominal cramping | Drug induced hypothermia |
|---|---|---|---|---|---|---|
| Paiteling | 319 | 113(35.4) | 268(84.0) | 248(77.7) | 179(56.1) | 4(1.0) |
| Interferon | 256 | 18(7.0) | 0 | 0 | 0 | 0 |
| $X^2$ | | 65.078 | - | - | - | - |
| P-value[a] | | <0.001 | - | - | - | - |

[a]P-value was calculated by chi-square test.

much higher than the cumulative effective rate of Interferon 41.4%. The results of this study also showed that the difference in the conversion rate of HR-HPV in 14 mono-infections between the two groups of drugs was statistically significant (p-value <0.001), with the conversion rate in the Paiteling group being significantly higher than that in the Interferon group. In HPV16, 18, 52, 56, and 68, the conversion rate of the Paiteling group was significantly better than that of the Interferon group, with P value <0.05, which suggests that Paiteling has better clinical efficacy against HPV.

A systematic review and Bayesian network meta-analysis study have shown that the application of various forms of vaginal administration, except for individual use of Lactobacilli vaginal capsules, is more efficacious than no treatment in patients with cervical persistent HR-HPV infection after excisional treatment [10]. A retrospective study aims to evaluate the therapeutic outcomes of Paiteling and CO2 laser therapy on high-risk human papillomavirus. The results of the study show that the percentage of persistent HR-HPV clearance rate for Paiteling patients was higher than carbon dioxide laser vaporization [11]. In addition, the results of a 2-year follow-up comparing HPV-positive with HPV-negative patients with Urothelial Carcinoma of the Bladder showed that the presence of HPV DNA was associated with a trend toward a higher recurrence rate [12]. During early HPV infection, HPV virus particles infect basal cells, the epidermis, through the broken epidermal mucosal cell layer, rarely involving the dermis of the skin. Clinical methods used to try to clear HPV include a variety of methods, such as antiviral gels, interferon, herbal preparations, physiotherapy, etc., of which interferon is commonly used in the treatment of HPV infection, its mechanism of action is to inhibit viral division and proliferation, in addition to stimulating the activity of T-cells as well as natural killer cells, inhibit the proliferation of epidermal cells, and enhance the phagocytosis of patients' macrophage cells, but previous studies showed that However, previous studies have shown that the effect of interferon therapy is not satisfactory [4]. In recent years, we have found that Paiteling inhibits and removes HPV infection mainly through its exfoliative effect on cervical epithelial cells, the metabolism rate of cervical epithelial cells in patients is significantly accelerated after the use of Paiteling. Yunhua Liu's team, by integrating LC-MS/MS, network pharmacology, and pharmacological experiments, has found that Paiteling can induce cervical cancer cell apoptosis by inhibiting the E6/E7-Pi3k/Akt signaling pathway [13]. This may provide an effective alternative strategy for traditional Chinese medicine in the treatment of epithelial neoplasias caused by HPV infection.

In conclusion, Paiteling, as a purely traditional Chinese medicine, has been observed to be efficacious against HR-HPV infection, reducing the risk of progression to cervical cancer in patients with HR-HPV, and the associated adverse effects are relatively few, most of which can gradually disappear after stopping the drug, making it a better therapeutic choice. However, Paiteling, as a pure Chinese medicine preparation, has not yet been included in the scope of medical insurance, and its cost is higher than that of interferon, which may increase the economic burden for patients who use it. The use of Paiteling treatment can avoid cervical physiotherapy and electrocution and other cervical damage operations, related side effects are small, operation repeatability is strong, at the same time, for the reproductive needs of patients can avoid the cervical structure of the cervix damage, reduce the patient's psychological and surgical costs, and thus, promoting the clinical application of these findings is highly recommended.

At the same time, this study also has certain shortcomings, such as the observation time is still short, whether the therapeutic effect of Paiteling on HPV infection is temporary, whether HPV will recur after treatment, and whether the long-term efficacy is not yet certain. This study was conducted on a single-center basis, so the conclusions may be unconvincing, and it is expected that large-sample, multi-center studies can be conducted in the future to increase the persuasiveness of the findings.

## Supporting information

**S1 Dataset.**
(XLS)

## Author Contributions

**Conceptualization:** Cunjian Yi.

**Funding acquisition:** Cunjian Yi.

**Investigation:** Na He.

**Methodology:** Wenrong He, Cunjian Yi.

**Resources:** Wenrong He.

**Software:** Wumei Jin.

**Supervision:** Qingsong Zeng, Wenrong He.

**Validation:** Wumei Jin.

**Visualization:** Na He, Lin Song, Qingsong Zeng.

**Writing – original draft:** Na He, Lin Song, Zhaoxia Lu.

**Writing – review & editing:** Na He, Lin Song, Zhaoxia Lu.

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
