## [Decision Letter · Decision Letter 0]

15 Oct 2024

PONE-D-24-38433Clinical efficacy analysis of the Chinese medicine Paiteling applied to human papillomavirus infection: a retrospective studyPLOS ONE

Dear Dr. Yi,

Thank you for submitting your manuscript to PLOS ONE. After careful consideration, we feel that it has merit but does not fully meet PLOS ONE’s publication criteria as it currently stands. Therefore, we invite you to submit a revised version of the manuscript that addresses the points raised during the review process.

We look forward to receiving your revised manuscript.

Kind regards,

Kazunori Nagasaka

Academic Editor

PLOS ONE

**Journal Requirements:**

2. In the online submission form, you indicated that Data are available from the corresponding author for researchers who meet the criteria for access to confidential data.

**Additional Editor Comments:**

Dear Authors,

Thank you very much for your submission to PLOS ONE. As both reviewers noted, the manuscript is intriguing. I recommend that you revise the manuscript in accordance with their comments.

I look forward to receiving your revised version.

Sincerely,

Kazunori Nagasaka

Reviewers' comments:

Reviewer's Responses to Questions

**Comments to the Author**

1. Is the manuscript technically sound, and do the data support the conclusions?

Reviewer #1: Yes

Reviewer #2: Yes

Reviewer #3: Partly

2. Has the statistical analysis been performed appropriately and rigorously? 

Reviewer #1: Yes

Reviewer #2: Yes

Reviewer #3: Yes

3. Have the authors made all data underlying the findings in their manuscript fully available?

Reviewer #1: Yes

Reviewer #2: Yes

Reviewer #3: Yes

4. Is the manuscript presented in an intelligible fashion and written in standard English?

Reviewer #1: Yes

Reviewer #2: Yes

Reviewer #3: No

5. Review Comments to the Author

**Reviewer #1:** Both in the interferon group and in the paiteling group, the complications related to use should be mentioned in more detail.

Especially the place of interferon therapy today is controversial.

Viral clearance occurs in 90% of these patients within 2 years. In this case, the effectiveness of treatment is questionable.

It would have been more effective if the authors had conducted this study in symptomatic HPV infection cases.

It is debatable how accurate it is only in HPV-DNA+ cases.

Nevertheless, it should be evaluated because it is an original study.

The authors can add the study below as a reference to increase the value of manuscript.

Sarier M, Usta SS, Turgut H, et al. Prognostic value of HPV DNA in Urothelial Carcinoma of the Bladder: A Preliminary Report of 2-Year Follow-up Results. Urol J. 2021;19(1):45-49. doi:10.22037/uj.v18i.6429

**Reviewer #2:** 1. Data completeness: the article states that no datasets were generated or analysed, which could mean that the study results were based solely on existing medical records without additional data collection to validate those records.

2. Sample size: Although the study included 575 patients with HPV infection, this number may still not be large enough for some subgroups to be analysed, especially when multiple HPV subtypes are considered.

3. Confounding factors: studies may not adequately consider or control for all the confounding factors that may affect the results, such as the patient's age, lifestyle, and allergies to the drug's ingredients。

4. Failure to mention the control group: The article did not mention whether there was a control group using placebo or other treatments, which is crucial for assessing the efficacy of Patulin.

5. Reporting of side effects: The article mentions that some patients in the PATLIN group experienced mild side effects, but these were mostly tolerable and may have disappeared on their own at the end of treatment. However, the reporting of side effects may not be comprehensive enough, as only serious or persistent side effects are more likely to be reported.

6. Statistical analysis: Although statistical software was used to analyse the data, the article does not detail whether the statistical tests used were best suited to the type of data or whether they were adjusted for all possible confounders.

7. Standardisation of treatment: The study mentions the treatment modalities of Patulin and Interferon, but does not detail whether the treatment was administered in exactly the same way in all patients, which may have affected the consistency of the results.

8. Failure to discuss economic costs: The article did not discuss the economic costs of the treatment with PATRIN, which is important for assessing its feasibility in the real world.

**Reviewer #3:** The study by He et al., titled “Clinical Efficacy Analysis of the Chinese Medicine Paiteling Applied to Human Papillomavirus Infection: A Retrospective Study,” evaluates the effectiveness of Paiteling in treating HPV infections on a population level. This retrospective study included 575 HPV-infected patients from the First People’s Hospital of Jingzhou Gynecology Outpatient Clinic, covering the period from October 2020 to December 2023. The patients were divided into two groups: 319 received Paiteling, while 256 were treated with interferon.

A key strength of this study is the large sample size in both treatment groups. However, a significant limitation is the absence of a clearly defined or sufficiently long post-treatment follow-up period to assess the efficacy of Paiteling compared to interferon. Although the study appears to combine both retrospective (HPV infection) and prospective (treatment and efficacy evaluation) elements, it is classified as a retrospective study. Furthermore, the authors did not clearly explain how this study differs from previous similar studies conducted in China, nor did they adequately compare their findings with earlier research, leading to a somewhat brief discussion.

Major and general comments:

-The authors did not evaluate factors influencing the effectiveness of Paiteling treatment, such as age, baseline viral load, HPV type, single versus multiple HPV infections, immunological and nutritional status, obesity, and other comorbidities.

-Additionally, details on the duration and number of post-treatment follow-ups, as well as the number of patients lost to follow-up, are missing.

-The discussion section is superficial and could benefit from comparisons with previous studies on the clinical efficacy of Paiteling treatment for cervical cancer and the clearance of persistent HPV infections. It is important to determine whether the current findings align with or differ from existing evidence on the effectiveness of Paiteling treatment, to highlight the significance of this study.

-The strengths and limitations of the study should also be addressed. Although the sample size is relatively large, the subcluster analysis of treatment outcomes by HPV types in the two intervention groups is limited by small sample sizes and lacks statistical power. Except for HPV types 16, 18, 52, and 58, the sample sizes for other high and low-risk HPV types are extremely low, which is a common limitation of retrospective study designs in clinical research.

Minor and specific comments:

-Line 90-91: What does the term “information” mean? It might refer to detailed patient information, including demographics, medical history, and clinical outcomes. If this was the case, of course, it was, the authors should delete information in “Information and Methods” and Replace “1.1 General Information” with “1.1 Study Subjects”.

-Line 96: The phrase “those who met the indications for colonoscopy” is unclear. It likely means patients who had medical reasons or criteria that warranted a colonoscopy. You could rephrase it to: “patients who had medical indications for undergoing a colonoscopy.”

-Line 98-99: This sentence “Five hundred and seventy-five patients who met the inclusion and exclusion criteria were screened …” appears incorrect. If the 575 subjects were enrolled because they met the inclusion criteria, you should delete “and exclusion”. The revised sentence would be: “Five hundred and seventy-five patients who met the inclusion criteria were screened.”

-Line 115: “1.3 Methods of Administration” appeared to me unclear. If it refers to how drugs were administered, you should specify this. You could rephrase it to: “1.3 Methods of Drug Administration”. Also, since this section was written as a proposal or protocol, and needs to be rewritten in the past tense. For example: “The drugs were administered according to the following methods…”

-Line 144: Including a figure that demonstrates the flow of participants through the study is a great idea. A figure titled “A flow chart illustrating the number of participants enrolled in different intervention groups, dropouts, and the number of individuals who cleared the virus versus those who did not” needs to be provided in the methods or result section for clarity of the manuscript.

-Line 177: Replace “see Table 2 for details” with “(Table 2)”.

-Line 209: The section on adverse reactions should be more detailed. I suggest generating a table that summarises the findings on adverse reactions. The table provides figures on the number of study participants who developed adverse reactions in the two intervention groups, along with the most commonly observed adverse reactions.

-Line 247: The term “scholars” is vague. It could be referring to researchers or experts. You might want to clarify this based on the context.

-Lines 263-264: The sentence “reduce the patient’s psychological and economic burden, it is worth to promote the clinical” seems incomplete. You could rephrase it to: “ ..reduce the patient’s psychological and economic, and thus, promoting the clinical application of these findings is highly recommended.”

6. PLOS authors have the option to publish the peer review history of their article (what does this mean?). If published, this will include your full peer review and any attached files.

Reviewer #1: No

Reviewer #2: **Yes: **Shuguang Zhou

Reviewer #3: **Yes: **Tesfaye Gelanew

---

## [Author Response · Author response to Decision Letter 0]

9 Nov 2024

Point-by-Point Response to the comments by Reviewers

Journal: PLOS ONE 

Manuscript ID: PONE-D-24-38433

Title: Clinical efficacy analysis of the Chinese medicine Paiteling applied to human papillomavirus infection: a retrospective study

Author(s): Na He, Lin Song, Zhaoxia Lu, Qingsong Zeng, Wumei Jin, Wenrong He, Cunjian Yi.

Corresponding authors: 

Name: Cunjian Yi, e-mail: 2022721105@yangtzeu.edu.cn

Wenrong He, e-mail: hewenrong@126.com

Dear editors and reviewers:

Thank you for offering us an opportunity to improve the quality of our submitted

manuscript entitled “Clinical efficacy analysis of the Chinese medicine Paiteling applied to human papillomavirus infection: a retrospective study ” (ID: PONE-D-24-38433 ). We appreciated very much the reviewers’ constructive and insightful comments. We have studied comments carefully and have made corrections which we hope meet with approval. Revise portions are marked in red in the revised manuscript. If there are any incorrect answers or questions in the manuscript, please do not hesitate to let us know. 

A Point-by-Point Response to the reviewer’s comments is given below. 

The detailed explanation is given as follows:

Reviewer #1 Summary of the revision:

1.Both in the interferon group and in the paiteling group, the complications related to use should be mentioned in more detail.

2. Especially the place of interferon therapy today is controversial.

3. Viral clearance occurs in 90% of these patients within 2 years. In this case, the effectiveness of treatment is questionable. It would have been more effective if the authors had conducted this study in symptomatic HPV infection cases. It is debatable how accurate it is only in HPV-DNA+ cases.

4. Nevertheless, it should be evaluated because it is an original study.

5. The authors can add the study below as a reference to increase the value of manuscript.

Sarier M, Usta SS, Turgut H, et al. Prognostic value of HPV DNA in Urothelial Carcinoma of the Bladder: A Preliminary Report of 2-Year Follow-up Results. Urol J. 2021;19(1):45-49. doi:10.22037/uj.v18i.6429

Responses to a reviewer’s comments. 

Thanks very much for your approval of this study, below are the replies to your

constructive questions：

1. Comment: Both in the interferon group and in the paiteling group, the complications related to use should be mentioned in more detail.

1. Reply: We feel great thanks for your professional review work on our article. As you are concerned, we believe that the adverse effects of both drug groups merit a more detailed discussion. According to your nice suggestions, we have made extensive corrections to our previous manuscript, the detailed corrections are listed below. 

Key points added in Section 2.5: “ The incidence of adverse reactions was higher in the Paiteling group than in the Interferon group, and several different side effects may occur during the same patient's dosing period. For example, patients may experience symptoms such as localized burning sensation and Shedding of cervical epidermal cells simultaneously. In contrast, only 18 patients in the interferon group reported an increase in conscious vaginal discharge. The specific data are detailed in Table 4.” Please check out sections 2.5 on line 205-212 for specific modifications, tks.

Table 4 Comparison of the occurrence of adverse reactions in the two groups [n (%)]

Groups Total cases Increased vaginal secretions Shedding of cervical epidermal cells Localized burning sensation Lower abdominal cramping Drug induced hypothermia

Paiteling 319 113(35.4) 268(84.0) 248(77.7) 179(56.1) 4(1.0)

Interferon 256 18(7.0) 0 0 0 0

X² 65.078 - - - -

P-value ＜0.001 - - - -

2. Comment: Especially the place of interferon therapy today is controversial.

2. Reply: Thank you for this valuable comment. Yes, There is currently some controversy in the international medical community regarding the efficacy and applicability of the use of interferon for the treatment of HPV infection. In the field of HPV treatment, neither international guidelines nor domestic expert consensus currently recommends any specific drug or treatment to completely cure HPV infection. Interferon, as a broad-spectrum antiviral drug, is still used by many doctors and patients in the clinic due to its advantages of low cost and broad applicability. Based on these considerations, our team chose interferon as a control group to evaluate its actual effect on HPV treatment.

3. Comment: Viral clearance occurs in 90% of these patients within 2 years. In this case, the effectiveness of treatment is questionable. It would have been more effective if the authors had conducted this study in symptomatic HPV infection cases. It is debatable how accurate it is only in HPV-DNA+ cases.

3. Reply: We were really sorry for our careless mistake. Thank you for your reminder. In response to your inquiry regarding the natural clearance rate of HPV within two years, we regret to acknowledge an oversight in our article that failed to clearly communicate that our study subjects were all patients with persistent infections. Specifically, the majority of patients in the Paiteling group were included in the study because their infections persisted despite conventional antiviral therapy. Key points added in Section 1.2: “Persistent HPV positivity and no cancer in colposcopic cervical biopsy...” Please check out sections 1.2 on line 104-105 for specific modifications, tks.

4. Comment: Nevertheless, it should be evaluated because it is an original study.

4. Reply: We appreciate your summary of the manuscript and encouraging comment. 

5. Comment: The authors can add the study below as a reference to increase the value of manuscript.

Sarier M, Usta SS, Turgut H, et al. Prognostic value of HPV DNA in Urothelial Carcinoma of the Bladder: A Preliminary Report of 2-Year Follow-up Results. Urol J. 2021;19(1):45-49. doi:10.22037/uj.v18i.6429

Reply: We sincerely appreciate the valuable comments. We have checked the literature carefully and added reference on “Prognostic value of HPV DNA in Urothelial Carcinoma of the Bladder: A Preliminary Report of 2-Year Follow-up Results.” into the DISCUSSION part in the revised manuscript. Please check out sections 3 on line 235-238 for specific modifications, tks.

Reviewer #2 Summary of the revision:

1. Data completeness: the article states that no datasets were generated or analysed, which could mean that the study results were based solely on existing medical records without additional data collection to validate those records.

2. Sample size: Although the study included 575 patients with HPV infection, this number may still not be large enough for some subgroups to be analysed, especially when multiple HPV subtypes are considered.

3. Confounding factors: studies may not adequately consider or control for all the confounding factors that may affect the results, such as the patient's age, lifestyle, and allergies to the drug's ingredients.

4. Failure to mention the control group: The article did not mention whether there was a control group using placebo or other treatments, which is crucial for assessing the efficacy of Patulin.

5. Reporting of side effects: The article mentions that some patients in the PATLIN group experienced mild side effects, but these were mostly tolerable and may have disappeared on their own at the end of treatment. However, the reporting of side effects may not be comprehensive enough, as only serious or persistent side effects are more likely to be reported.

6. Statistical analysis: Although statistical software was used to analyse the data, the article does not detail whether the statistical tests used were best suited to the type of data or whether they were adjusted for all possible confounders.

7. Standardisation of treatment: The study mentions the treatment modalities of Patulin and Interferon, but does not detail whether the treatment was administered in exactly the same way in all patients, which may have affected the consistency of the results.

8. Failure to discuss economic costs: The article did not discuss the economic costs of the treatment with PATRIN, which is important for assessing its feasibility in the real world.

Responses to a reviewer’s comments. 

Thanks very much for your approval of this study, below are the replies to your

constructive questions：

1. Comment: Data completeness: the article states that no datasets were generated or analysed, which could mean that the study results were based solely on existing medical records without additional data collection to validate those records.

1. Reply: I agree with you, thanks very much for your advice. Our data were primarily derived from existing medical records and follow-up information, and we have not yet performed additional validation with external data. However, the points you raised are indeed very pertinent, and we strongly believe that incorporating your suggestions into future studies will greatly enhance the credibility of our findings.

2. Comment: Sample size: Although the study included 575 patients with HPV infection, this number may still not be large enough for some subgroups to be analysed, especially when multiple HPV subtypes are considered.

2. Reply: We gratefully appreciate for your valuable comment. Currently, our data collection is retrospective, and as such, we have been able to gather limited patient information. Some patients who were lost to follow-up or failed to undergo timely reviews were not included in our study. To enhance the credibility of our findings, we are considering conducting a multicenter study with a large sample size in the future. We believe that by expanding our research scope and deepening our data analysis, we can significantly increase the reliability and persuasiveness of our results. Your suggestions are greatly appreciated, and we are committed to striving for higher research standards.

3. Comment: Confounding factors: studies may not adequately consider or control for all the confounding factors that may affect the results, such as the patient's age, lifestyle, and allergies to the drug's ingredients.

3. Reply: We value your professional comments on our article. In response to your question about confounders, we did face some challenges in our study. Our study was based on a retrospective collection of data from patients who attended the gynecology outpatient clinic of the First People's Hospital of Jingzhou City between October 1, 2020, and December 1, 2023, retrospectively. Due to the nature of retrospective studies, our study had some limitations, especially in the absence of detailed information about patients' lifestyles and drug allergy histories, among others, which limited our ability to control for relevant confounders. We recognize this and will strive to overcome these limitations in future studies to provide a more comprehensive and precise analysis. Thank you again for your valuable input, which is essential for us to improve our research methods.

4. Comment: Failure to mention the control group: The article did not mention whether there was a control group using placebo or other treatments, which is crucial for assessing the efficacy of Patulin.

4. Reply: We sincerely appreciate the valuable feedback you provided on our article, and your suggestions are crucial in helping us improve the quality of our manuscript. Regarding your reference to the control group, we did not set up a blank group but added only the interferon group as a control. This decision was primarily due to the incomplete case data for some unmedicated patients and the difficulties encountered during follow-up. Nonetheless, we have introduced data from other research teams in the discussion section that included studies with blank control groups. The results showed that the HPV conversion rate was higher in both Paiteling and Interferon groups compared to the naturally observed group. We will continue to monitor the latest research in this area and consider a more comprehensive control group setup in our subsequent work. Thank you again for your valuable comments.

5. Comment: Reporting of side effects: The article mentions that some patients in the PATLIN group experienced mild side effects, but these were mostly tolerable and may have disappeared on their own at the end of treatment. However, the reporting of side effects may not be comprehensive enough, as only serious or persistent side effects are more likely to be reported.

5. Reply: Thank you very much for your valuable comments. According to your nice suggestions, we have made corrections to our previous manuscript, the detailed corrections are listed below. Key points added in Section 2.5: “ The incidence of adverse reactions was higher in the Paiteling group than in the Interferon group, and several different side effects may occur during the same patient's dosing period. For example, patients may experience symptoms such as localized burning sensation and Shedding of cervical epidermal cells simultaneously. In contrast, only 18 patients in the interferon group reported an increase in conscious vaginal discharge. The specific data are detailed in Table 4.” Please check out sections 2.5 on line 205-212 for specific modifications, tks.

We have not yet determined if there are serious or persistent side effects from using Paiteling during the observation period. We will extend the follow-up period and enhance the thoroughness of follow-up to ensure the safety of patients' use of the drug. However， no severe side effects were reported by the Paiteling patients. This is due to the fact that Paiteling, as a traditional Chinese medicine, is a topical, non-invasive medicine, thus preserving the integrity of the cervix.

6. Comment: Statistical analysis: Although statistical software was used to analyse the data, the article does not detail whether the statistical tests used were best suited to the type of data or whether they were adjusted for all possible confounders.

6. Reply: We thank the reviewer of pointing out this issue. We should indeed enhance the presentation of statistics with more detail. We have revised in the section 1.5: “In this study, the data were analyzed and processed using the statistical software SPSS version 25.0. The Mann-Whitney U test was used to assess the age-related variance in the measurement data, and the chi-square test was used to compare the rates of categorical data. Results were considered statistically significant if P < 0.05.” Thanks for your reminder. Please check section 1.5 on line 132-135, tks.

7. Comment: Standardisation of treatment: The study mentions the treatment modalities of Patulin and Interferon, but does not detail whether the treatment was administered in exactly the same way in all patients, which may have affected the consistency of the results.

7. Reply: Thanks to the review for this careful comment. Both groups of patients involved in this study received a standardized course of treatment based on hospital guidelines and medication recommendations, ensuring that each patient received the same treatment.

8. Comment: Failure to discuss economic costs: The article did not discuss the economic costs of the treatment with PATRIN, which is important for assessing its feasibility in the real world.

8. Reply: We agree with the reviewer’s suggestion and will incorporate the recommended changes into the manuscript. Key points added in Section 3:“However, Paiteling, as a pure Chinese medicine preparation, has not yet been included in the scope of medical insurance, and its cost is higher than that of interferon, which may increase the economic burden for patients who use it.” Thanks for your reminder. Please check section 3 on line 260-263, tks.

Reviewer #3 Summary of the revision:

Major and general comments:

1.The authors did not evaluate factors influencing the effectiveness of Paiteling treatment, such as age, baseline viral load, HPV type, single versus multiple HPV infections, immunological and nutritional status, obesity, and other comorbidities.

2.Additionally, details on the duration and number of post-treatment follow-ups, as well as the number of patients lost to follow-up, are missing.

3.The discussion section is superficial and co

---

## [Decision Letter · Decision Letter 1]

26 Nov 2024

Clinical efficacy analysis of the Chinese medicine Paiteling applied to human papillomavirus infection: a retrospective study

PONE-D-24-38433R1

Dear Dr. Yi,

We’re pleased to inform you that your manuscript has been judged scientifically suitable for publication and will be formally accepted for publication once it meets all outstanding technical requirements.

Kind regards,

Kazunori Nagasaka

Academic Editor

PLOS ONE

Additional Editor Comments (optional):

Dear Authors,

Thank you for submitting your manuscript to Plos One.

Our reviewers have recommended that the manuscript is ready for publication. Congratulations on your work! We look forward to receiving your future studies.

Sincerely,

Kazunori Nagasaka

Reviewers' comments:

Reviewer's Responses to Questions

**Comments to the Author**

1. If the authors have adequately addressed your comments raised in a previous round of review and you feel that this manuscript is now acceptable for publication, you may indicate that here to bypass the “Comments to the Author” section, enter your conflict of interest statement in the “Confidential to Editor” section, and submit your "Accept" recommendation.

Reviewer #1: All comments have been addressed

Reviewer #3: All comments have been addressed

2. Is the manuscript technically sound, and do the data support the conclusions?

Reviewer #1: Yes

Reviewer #3: Yes

3. Has the statistical analysis been performed appropriately and rigorously? 

Reviewer #1: Yes

Reviewer #3: Yes

4. Have the authors made all data underlying the findings in their manuscript fully available?

Reviewer #1: Yes

Reviewer #3: Yes

5. Is the manuscript presented in an intelligible fashion and written in standard English?

Reviewer #1: Yes

Reviewer #3: Yes

6. Review Comments to the Author

Reviewer #1: Comments are addressed this paper is ready to publish

Reviewer #3: I have no more comments! However, authors need to meticulously revise to minimize grammatical errors here and there. They need to expand and discuss the limitations of the study, given it has several limitations.

7. PLOS authors have the option to publish the peer review history of their article (what does this mean?). If published, this will include your full peer review and any attached files.

Reviewer #1: No

Reviewer #3: **Yes: **Tesfaye Gelanew

---

## [Editor Report · Acceptance letter]

29 Nov 2024

PONE-D-24-38433R1 

PLOS ONE

Dear Dr. Yi, 

I'm pleased to inform you that your manuscript has been deemed suitable for publication in PLOS ONE. Congratulations! Your manuscript is now being handed over to our production team.

Kind regards, 

on behalf of

Professor Kazunori Nagasaka 

Academic Editor

PLOS ONE